# Deep Scattering Spectrum Germaneness for Fault Detection and Diagnosis for Component-Level Prognostics and Health Management (PHM)

**DOI:** 10.3390/s22239064

**Published:** 2022-11-22

**Authors:** Ali Rohan

**Affiliations:** 1Department of Mechanical, Robotics and Energy Engineering, Dongguk University, 30 Pildong 1 Gil, Jung-gu, Seoul 04620, Republic of Korea; ali_rohan2003@hotmail.com or ali.rohan@nottingham.ac.uk; 2Faculty of Medicine and Health Sciences, University of Nottingham, Sutton Bonington, Loughborough LE12 5RD, UK

**Keywords:** deep scattering spectrum, wavelet scattering network, prognostics and health management (PHM), fault detection and diagnosis

## Abstract

Most methodologies for fault detection and diagnosis in prognostics and health management (PHM) systems use machine learning (ML) or deep learning (DL), in which either some features are extracted beforehand (in the case of typical ML approaches) or the filters are used to extract features autonomously (in the case of DL) to perform the critical classification task. In particular, in the fault detection and diagnosis of industrial robots where the primary sources of information are electric current, vibration, or acoustic emissions signals that are rich in information in both the temporal and frequency domains, techniques capable of extracting meaningful information from non-stationary frequency-domain signals with the ability to map the signals into their constituent components with compressed information are required. This has the potential to minimise the complexity and size of traditional ML- and DL-based frameworks. The deep scattering spectrum (DSS) is one of the approaches that use the Wavelet Transform (WT) analogy for separating and extracting information embedded in a signal’s various temporal and frequency domains. Therefore, the primary focus of this work is the investigation of the efficacy and applicability of the DSS’s feature domain relative to fault detection and diagnosis for the mechanical components of industrial robots. For this, multiple industrial robots with distinct mechanical faults were studied. Data were collected from these robots under different fault conditions and an approach was developed for classifying the faults using DSS’s low-variance features extracted from input signals. The presented approach was implemented on the practical test benches and demonstrated satisfactory performance in fault detection and diagnosis for simple and complex classification problems with a classification accuracy of 99.7% and 88.1%, respectively. The results suggest that, similarly to other ML techniques, the DSS offers significant potential in addressing fault classification challenges, especially for cases where the data are in the form of signals.

## 1. Introduction

A significant proportion of tasks in modern industries are performed by devices such as robots. These robots are composed of a variety of components that work together to perform a specially designed operation. Due to their continuous operation, these components are prone to deterioration, which most of the time can lead to fatal damage. Therefore, it is necessary that over a certain period, proper assessment and maintenance strategies are in place. To help tackle this, prognostics and health management (PHM) has emerged as an appealing method for developing methodologies for system health monitoring, diagnostics, RUL prediction, and prognostics. PHM is regarded as an efficient method that is capable of delivering complete, accurate, and customized solutions for system health management [1]. PHM has three primary functions. The first and foremost is the early detection of the fault triggers in a system, followed by the separation and recognition of faults and their respective source, and finally predicting the remaining useful life (RUL) of a specific component or a system. Figure 1 depicts the fundamental tasks performed by a PHM system. PHM can be used at the component, system, or both levels. PHM at the component level aims to create health monitoring methodologies for electromechanical components such as electric motors, electronic devices, bearings, gear reducers, etc. It evaluates if the monitored component’s health deteriorated over time because of numerous environmental, operational, and performance-related parameters [2,3]. However, PHM at the system level evaluates detailed system health, considering system operations, designs, and process-related parameters [4].

The most essential of the three PHM tasks is fault detection and diagnosis. The sustainability of a PHM system is heavily reliant on it. If the fault is not adequately diagnosed, the entire PHM system may fail. There has been a great development in the fault detection and diagnosis of a PHM system in recent years, with both mathematical model-based [5,6] and data-driven [7,8] techniques yielding encouraging results. Data-driven techniques, in particular, are gaining prominence due to their capacity to adapt and transform in real-time under diverse conditions. Furthermore, there have been significant advancements in the computing capability of devices with enhanced sensor technologies that allow for effective data acquisition. These techniques are mostly based on machine learning (ML) and deep learning (DL), in which handcrafted features or deep neural networks with numerous layers of back-to-back filters are used to learn important information in a given dataset. To develop a PHM system using current data-driven methodologies, considerable information and additional real-time data, such as vibration, acoustic emission, laser displacement, temperature, speed, and electric current [9,10,11,12,13], are required.

Previously, several methods have been able to retrieve statistical characteristics from audio data and categorized them using standard machine learning algorithms. The support vector machine (SVM) and artificial neural network (ANN) classifiers were trained using six different statistical features such as range, mean, standard deviation, kurtosis, skewness, and crest factor [14]. A method for autonomous mining operations based on vibration signals was developed in [15]. The authors employed a low-pass Butterworth filter to process and analyze vibration signals before extracting eight time-domain features, eight frequency-domain features, and five Morlet wavelet features. A technique using Mel-frequency cepstrum eoefficients (MFCCs) from audio signals with SVM for classifications was presented in [16]. The use of MFCCs features and their comparison with a library of features to determine if a bike engine is healthy or malfunctioning was proposed in [17]. In contrast, Ref. [18] developed an ML-based fault identification and diagnosis technique for electric current signals based on a distinct feature selection, extraction, and infusion procedure.

For DL, Ref. [19] created a diagnosis technique for the health status characterization of the centrifugal pump faults based on sensory data fusion and a convolutional neural network (CNN). Ref. [20] described a similar methodology for sensor fault diagnosis, in which a computer software was used to generate defective sensor data, and the signal recognition task was turned into an image processing task using CWT once incorrect data were generated. CNN was used to identify the sensor faults. Recent research studies achieved impressive results when employing visual representations of audio signals to train cutting-edge deep learning architectures such as CNNs for the problem of acoustic emission machine faults [21,22,23,24]. Researchers in [25,26] presented tool wear condition detection methods based on deep learning with multi-cutting force time series signals and an entropy-based sparsity measure for the prognosis of bearing defects.

The techniques described above are effective for performing fault classification tasks for various fault kinds in a variety of settings. However, the majority of these techniques have some distinct disadvantages. Traditional handcrafted feature extraction approaches need a specialist understanding of the type of input data. In general, the input data for PHM systems consist of vibration or acoustic emission signals [27]. These signals are non-stationary and might differ from one fault diagnostics application to another. As a result, generalizing a feature space becomes a highly complicated issue. In most cases, the features are only applicable to a single component’s failure. The DL-based algorithms, on the other hand, provide leverage over the manual and labor-intensive processes of feature extraction. At various layers, feature extraction is accomplished autonomously using filters with varying kernel sizes. However, developing the architecture of an appropriate deep neural network model remains a time-consuming effort. The manual hit and trial approach is typically used to determine the number of filters, kernel size, number of layers, and hyper-parameters tuning. In certain instances, a certain number of layers may be sufficient, albeit it may be as computationally expensive for another task where fewer layers might yield promising results. Especially when it comes to the classification of faults where input data are composed of non-stationary signals, an intermediate feature extraction tool such as Fast Fourier Transform (FFT) [28,29] focusing on time continuum signals and frequency analysis, short-time Fourier transform (STFT) [30] focusing on time-frequency domain analyses or wavelet transform (WT) [31,32] is intensively used. These tools are used to generate a map of time-frequency domain features in a scalogram or spectrogram image, which is then inputted into a deep neural network, such as CNN. As a result, the computational challenges are exacerbated by one step. To overcome the aforementioned disadvantages, we investigate the use of the deep scattering spectrum (DSS), especially the scattering transform, for fault detection and the diagnosis of mechanical components of industrial robots in this work. DSS is specially developed for signal classification. It builds a deep network using the WT idea by performing a scattering transform with fixed settings of the dilated filter. It combines the strength of traditional signal-processing methods with the depth of a deep neural network. According to our understanding, scattering transforms in the literature are largely focused on audio applications, but a generic scattering representation for classification that applies to numerous signal modalities other than audio remains understudied [33,34]. For component-level PHM, we propose the application of the scattering transform to electric current signals, i.e., motor current signature analysis (MCSA), rather than audio, acoustic emission, or vibration data. MCSA outperforms vibration and acoustic emission analyses in various ways. MCSA uses the inherent current signal of the motor control unit, requiring no extra sensors, that reduces costs and system complexities. Furthermore, the current signals are distinct and not easily influenced by adjacent working conditions. Lastly, earlier solutions used balanced datasets that are not easily available in the context of industrial robots. By expanding this study, we implement DSS with simple classifiers for an imbalanced scarce and multi-domain (ISMD) dataset.

The details of this study are presented in the following sections. Section 2 defines the materials and methods used in this study, including the experimental test bench and descriptions of the suggested technique. Section 3 consists of the results and discussions, and Section 4 presents the conclusion.

## 2. Materials and Methods

Figure 2 depicts the fundamental process of the approach used in this work. Multiple industrial robot-based experimental setups were utilized to obtain real-time data by inducing mechanical component faults. The acquired data were pre-processed to remove ambient noise, and signal segmentation was used to divide the raw recorded signal into successive segments based on the robots’ motion patterns. The segmented signals were then processed with the scattering transform to extract the scattering and scalogram coefficients, resulting in a feature vector that was utilized to identify the faults. The specifics of the procedures are provided in the following subsections.

### 2.1. Experimental Setups

We concentrated on the practical implementation of the fault detection and diagnosis system in this study, taking into account the experimental setups depicted in Figure 3 and Figure 4. We validated the applicability of the given methodology for diverse industrial equipment settings using experimental setups for two distinct types of robots. The first experimental setup comprised three major parts: an industrial robot, a programmable logic controller (PLC), and a command-generating module. Robostar Co. (Ansan, Republic of Korea) manufactured the robot utilized in this setup, and the model number is R004. The second experimental setup, similarly to the first, is made up of three major components: an industrial robot, a controller, and a personal computer (PC). Hyundai Robotics Co. (Daegu, Republic of Korea) manufactured the robot utilized in this setup, and the model number is YS080. It can carry a maximum payload of 80 kgf. This robot is much larger than Robostar. The robots in both experimental setups have six axes or joints, each of which is designed with a different type of electric motor, allowing the robots to move freely around each axis. To increase or decrease rotation speeds, the motors are connected to reducers at each axis. The Hyundai robot is controlled by sending commands to the controller through a PC, but the Robostar employs a manual command-generating module that delivers operation commands to the PLC, which then controls electric motors to generate a certain motion. On each axis of each robot, three-phase servo motors are employed. The motors’ power is set to be dependent on the amount of mechanical load on each axis. Figure 5 shows the details of the robots, which are as follows: (a) Robostar R004 and (b) Hyundai Robot YS080.

### 2.2. Data Acquisition

We recorded single-phase electric current signals at each axis of the robot for the Robostar experimental setup, with each of the six-axes motors having current sensors installed. Hall-effect-based linear current sensors WCS6800 were employed as current sensors. Data were collected for two classes: *normal* and *faulty*. The fault replicated in this experimental setup was related to the strain wave gear reducer, which is a sort of mechanical gear system that employs a pliable spline with external teeth that is distorted by a spinning elliptical plug to connect with the internal gear teeth of an outer spline. Because of its compactness, lightweight feature, high gear ratio, and high torque capabilities, it is widely employed in robotic systems. The fault was mimicked by deforming the teeth of the internal gears in the gear reducer of the third-axis motor. The strain wave gear reducer is shown in Figure 6 for both the *normal* and *faulty* situations.

In the Hyundai robot experimental setup, motor current data were collected using current sensors for each of the three phases of the electric motor. The current sensors are mounted on each phase of the electric motors, resulting in a total of 18 current sensors to acquire 18 current signals for six electric motors. NI DAQ 9230 devices are used to record the current signals for each axis motor. This data acquisition device transmits the collected data to a PC executing LabView. The acquired signals are processed and a comprehensive database is created, containing the sensed data for each axis. Data are collected concurrently for each motor in various faulty conditions. In the first case, a Rotate vector (RV) reducer eccentricity-bearing fault was introduced into the reducer that was linked to the 4th axis motor. The fault was introduced in the second case by substituting the RV reducer with an aged/damaged one. The data were collected for three different classes: *normal*, *faulty* (RV reducer eccentric bearing fault), and *faulty aged* (RV reducer aging fault). The *faulty* and *faulty aged* RV reducers can be seen in Figure 7. Both robots were designed to work in all orientations along every axis of the rotation for several cycles. One cycle denotes the conclusion of a single range of motion across a single axis. For each axis, data were collected for ten cycles. Figure 8 illustrates the fundamental block diagram of the data collection procedure for a single-axis motor. It should be noted that the same procedure is performed for Robostar, with the exception that only electric current signals for a single-phase are collected to study the influence of different phases on fault classifications.

Subsequently, the motors were executed at various speeds ranging from 10% to 100% of their rated speed to see how the speed variation affected the PHM system. It is also performed to produce an imbalanced and multi-domain dataset. Figure 9 and Figure 10 display the equipment utilized in the data collection procedure for the Robostar and Hyundai robots. The electric currents are collected for each axis motor, even if the fault is only put into the gear reducer of a single axis because a fault in only one axis may impact the performance and effectiveness of some other axes motors due to mechanical linkages.

### 2.3. Data Pre-Processing

Unlike mathematical-model-based approaches, data-driven approaches are heavily impacted by how data are fed to a feature extractor, whether manual or automatic feature extractions. The effectivenss of a machine learning (ML) or deep learning (DL) method may be greatly enhanced if the data are pre-processed on the fly to eliminate superfluous information, redundancy, and undesirable features. Data preparation may not be essential in the case of image data since the image has a compact two-dimensional structure rather than a one-dimensional signal space. Because of the participation of multiple practical components existing in a single location, the signals, whether audio or electric current, tend to be noisy. The recorded signal data are extremely susceptible to ambient noise, redundancy, and superfluous information produced by the recording process, especially in the case of industrial robots with many electromechanical components conjoined to perform a particular task. Furthermore, the information must be organized in such a way that the computational complexity is maintained to a bare minimum at all times. We conducted various operations to eliminate ambient noise and superfluous information from the raw recorded electric current signals for this purpose. Unlike acoustic emission methods, which are commonly used to detect mechanical component faults, the motor current signature analysis (MCSA) provides leverage in safeguarding the original form of the signal. When compared to audio signals, electric current signals are less susceptible to noise.

Figure 11 shows a raw recorded electric current signal for Robostar when it is continuously operating along the faulty strain wave gear reducer axis. The fault is mimicked on axis 3 in the case of Robostar. The electric current signals were recorded for 110 s at a sampling rate of 2048 Hz at various speed ranges, from 10% to 100%. The signal shown in Figure 11 is for a 10% speed scenario and class: *normal*. The raw input signal was pre-processed by first eliminating the ambient noise with Savitzky–Golay filtering. Digital smoothing polynomial filters or least-squares smoothing filters are used in this case. Savitzky–Golay filters outperform typical averaging finite impulse response (FIR) filters, which tend to remove high-frequency components with noise and are optimum in the notion that they reduce the least-squares error in matching a polynomial to frames of the noisy dataset [35,36]. Following the elimination of silent sections from the signal, the resulting signal is divided into cycles of rotations. The rotation cycle is the movement of the robot in a direction along a single axis. A cycle is defined as two clockwise movements from the origin to a maximum range of motion and two anticlockwise movements back to the origin of the robot along one axis. The purpose of signal segmentation is to reduce the computational complexity of an ML or DL model. We used the idea of image dilation [37,38] rather than the traditional envelope detection approach to detect the envelope of the recorded signal since the traditional envelope detection technique failed to detect the changing envelope of the recorded signals in real time. Using image-dilation for envelope detection is beneficial as it successfully produces a spectrum of the boundary around the peaks of the signal regardless of the change in shape or sampling frequency. Figure 11 identifies the detected envelope in red color. For signal segmentation, we defined a threshold value (shown in Figure 11 in green color), which in this case is the median value, and compared it to the signal envelope to find the points of the actual recurrence of the cycle. Based on these points, the cycles were segmented and placed in a concatenated database. Figure 12 shows the general block architecture of the segmentation method, and Figure 13 features an example of segmented cycles for the signal shown in Figure 11. Note that the original recorded current signal contained 10 cycles whereas Figure 13 only shows 5 cycles as an example. The same procedure was adopted for the case of the Hyundai robot for data pre-processing for the classes of *normal*, *faulty*, and *faulty aged*.

### 2.4. Deep Scattering Spectrum

Deep Scattering Spectrum (DSS) has a similar architecture to a Deep Neural Network (DNN) where different levels of numerous filters are utilized to extract features from input data. Unlike the DNN, which uses a customizable dilation method to extract information from filters, DSS uses a fixed set of dilated filters. These filters are pre-defined, adhering to the wavelet transform’s (WT) characteristics. The scattering transform (ST) or wavelet scattering transform (WST) is the DSS’s core component. The WST is based on latent wavelets, providing consistent and stable informative signals from the input data set. Distortion tolerant, WST preserves class discriminability, making it suitable for classification. To acquire the signal characteristics, a sequential multi-wavelet decomposition based on modular arithmetic and local averaging is used. Multi-scale complex wavelets are used to acquire the low-frequency characteristics of the signals. High-frequency coefficients, on the other hand, are calculated, while relatively stable frequency characteristics are obtained by local averaging. The second high-frequency convolution wavelet is used to recover high-frequency information that has been lost as a result of the local averaging required to ensure the robustness of higher frequency terms. WST works with data in phases, which means the output of one phase is used as the input for the next. Each phase consists of three operations: (1) convolution, (2) non-linearity (modulus), and (3) averaging (scaling function); this is comparable to the convolutional neural network approach (CNN). Similar procedures are performed at each layer of the network in CNN; however, they are referred to as (1) convolution, (2) ReLU (rectification linear unit), and (3) max pooling. WST has a clear advantage over CNN because the former employs a fixed set of dilated filters or wavelets, and all features are typically compacted in the form of scattering coefficients within the network’s two levels or layers due to the wavelet filters’ high effectiveness in decomposing frequency domain information into substituent levels. Through the different scattering pathways, the scattering coefficients balance the data invariance and discriminations. WST resembles the physiological models of the cochlea and the auditory system [39,40], which are also employed in audio processing [41]. Furthermore, the DSS architecture is typically designed based on the properties of the input signal, such as the length of the signal; the scaling or averaging function, which can also be determined by determining the frequency spectrum of the signal; and the number of wavelet filters, which typically range from 1 filter per octave to a maximum of 32 or 64 filters per octave. The wavelet decomposition and WST provide an obvious advantage over conventional DNN for any application of signals where the essential information is in the frequency domain. As mentioned previously in the Introduction, WST in the literature is largely focused on audio applications, but a generic scattering representation for classification that applies to numerous signal modalities other than audio is still understudied, especially the applications where the signal type is an electric current. In contrast to audio, electric current signals contain a repetitive pattern in both the time and frequency domains. The information in the signal is highly correlated in the frequency spectrum, hence making it an ideal signal type for the application of WST.

Figure 14 is a visual representation of the WST procedure of this study, including the architecture of the network for two levels for the 1D signal. Where *m* represents the first-order, second-order, and third-order scattering transforms, *f* represents the input 1D signal, ∗ represents the convolutional operator, ϕ(t) represents the low pass filter, J represents the scale, ψ(t) represents the wavelet function, and Λ represents the family of wavelet indices. Mathematically, if $f\left(t\right)$ is the signal to be considered for WST, the wavelet function ψ and the low-pass filter ϕ are designed to create filters that cover all frequencies in a signal. Local invariant feature *f* is generated by convolution functions $\;S_{0}f\left(t\right)=f\phi_{J}\left(t\right)$, and high frequencies can be calculated as follows.
(1)$$\begin{array}{c} \begin{array}{c} |W_{1}|f=\{S_{0}f\left(t\right),|f*\psi_{j1}\left(t\right){\left|\right\}}_{j_{1}\in \wedge_{1}} \end{array} \end{array}$$

By averaging the wavelet coefficient, the first-order scattering coefficient can be calculated as follows:(2)$$\begin{array}{c} \begin{array}{c} S_{1}f\left(t\right)=\{|f*\psi_{j1}{\left|*\phi_{j}\left(t\right)\right\}}_{j_{1}\in \wedge_{1}} \end{array} \end{array}$$
where $S_{1}f\left(t\right)$ can be considered as a low-frequency component of $|f*\psi_{j1}|$; however, the high-frequency component can be extracted as follows.
(3)$$\begin{array}{c} \begin{array}{c} |W_{2}||f*\psi_{j1}|\;={\left\{S_{1}f\left(t\right),\;\left||f*\psi_{j1}|*\psi_{j_{2}}\left(t\right)\right|\right\}}_{j_{2}\in \wedge_{2}} \end{array} \end{array}$$ The second-order scattering coefficient can be obtained as follows.
(4)$$\begin{array}{c} \begin{array}{c} S_{2}f\left(t\right)={\left\{\left||f*\psi_{j1}|*\psi_{j_{2}}\right|*\phi_{j}\left(t\right)\right\}}_{j_{i}\in \wedge_{i}},\;i=1,2 \end{array} \end{array}$$ The wavelet modulus convolutions can be obtained by iterating the above process.
(5)$$\begin{array}{c} \begin{array}{c} \begin{array}{c} U_{n}f\left(t\right)={\left\{\left|\left||f*\psi_{j1}|*.....\right|*\psi_{j_{n}}\right|\right\}}_{j_{i}\in \wedge_{i}},\;\\ \;\;\;\;\;\;\;\;i=1,2,\ldots ,n. \end{array} \end{array} \end{array}$$ The *n*th order scattering coefficient $\left(S_{n}f\left(t\right)\right)$ can be obtained by averaging all the wavelet modulus convolutional coefficients ${(U}_{n}f\left(t\right))$ as follows.
(6)$$\begin{array}{c} \begin{array}{c} \begin{array}{c} S_{n}f\left(t\right)={\left\{\left|\left||f*\psi_{j1}|*.....\right|*\psi_{j_{n}}\right|*\phi_{j}\left(t\right)\right\}}_{j_{i}\in \wedge_{i}},\\ \;\;\;\;\;\;\;\;i=1,2,\ldots ,n. \end{array} \end{array} \end{array}$$ The final scattering matrix can be represented as follows.
(7)$$\begin{array}{c} \begin{array}{c} Sf\left(t\right)={\left\{S_{m}f\left(t\right)\right\}}_{0\leq n\leq l} \end{array} \end{array}$$

In Equation (Equation 7), l denotes the maximal decomposition order, and the final scattering matrix is calculated using features collected from each level of decomposition along several paths. The extracted features, in the form of a feature vector, are then utilized by a classifier to categorize the classes. In this paper, we used various classifiers to validate the efficacy of the WST’s feature domain. The section that follows contains information on the classifiers that were employed as well as the classification results.

## 3. Results and Discussion

Figure 15 and Figure 16 illustrate the electric current signal cycles for the Robostar and Hyundai robot experimental setups, respectively. Because each robot has its payload and operational parameters, the pattern of the current signals varies. Since the Robostar is a smaller robot, the electric motors used are of smaller specifications than those used in the Hyundai robot. As a result, the current amplitude during complete operations along a single axis is significantly lower than in the Hyundai robot. We present results and experiments from two types of problems. The first problem is for two classes—*normal* and *faulty* for the Robostar—whereas the second problem is for a complex classification problem created by the inclusion of the *faulty aged* class among the *normal* and *faulty* classes of the Hyundai robot due to the relevance of the *faulty aged* class with the *faulty* one. This is because, in the *faulty aged* class, the reducer gear is merely replaced with an old worn-out reducer. Secondly, the Hyundai robot’s size, capacity, and component structure render classification a considerably more challenging task. This can also be seen in Figure 16, where the electric current signals for a single rotational cycle tend to follow a fairly similar pattern across classes. The difference may still be seen, although not as clearly as in the case of the Robostar, where there is a noticeable difference in the signal amplitude and frequency domain.

These current signals were recorded at a sampling frequency of 2048 Hz and down-sampled to produce a comparable signal length to compare the outcomes for the same wavelet scattering network. The scattering network employed comprises two layers. For the scattering filter banks, the first layer had a quality factor of 8 while the second layer had a quality factor of 1. The number of wavelet filters per octave is the quality factor for each filter bank. The wavelet transform factorizes the scales by deploying the provided number of wavelet filters. The invariance scale, another important parameter in the architectural design of a wavelet scattering network, was computed using sampling frequency $f_{s}$ and the length of the input signal *N*. It is given in Equation (Equation 8).
(8)$$\begin{array}{c} \begin{array}{c} Invariance\;Scale\;=\;\frac{N/f_{s}}{2} \end{array} \end{array}$$

When the invariance scales are specified, the network becomes invariant to translations up to the invariance scale. The extent of the invariance in time or space is determined by the scaling function’s support. The scaling function and the coarsest-scale wavelet plot for the wavelet scattering network are shown in Figure 17. The invariance scale is computed using Equation (Equation 8), and based on the signal’s temporal domain, a scale of 1.6 s was chosen. The distances of the central frequencies of the wavelets in the filter banks are likewise affected by the invariance scale. However, in a scattering network, a wavelet’s temporal support cannot exceed beyond the invariance scale. The coarsest-scale wavelet plot exemplifies this feature. Frequencies less than the invariant scale are linearly separated, with the scale maintained constant such that the invariant’s size is not surpassed. Figure 18 depicts the wavelet scattering network’s filter banks at each layer. The first layer features eight filters per octave, whereas the second uses a single filter per octave. These filters are spread over the sampling frequency range to convolve and extract the features for the entire frequency spectrum of the signal.

Figure 19 illustrates the network path design based on the invariance scale, signal input length, and sampling frequency. As previously stated, the network contains two filter banks. The scattering paths are comprised of wavelets from both the first and second filter banks. The wavelet number and filter bank level of each wavelet filter on at least one path are indicated on the associated node. The network architecture consists of 329 paths, each containing 53 wavelets in the first layer and 8 wavelets in the second layer. The features at each layer are extracted using Equation (Equation 7), and a final feature vector consisting of 336 data features for each dataset for the Robostar and Hyundai robot is generated. These feature vectors were then used to classify faults using various sorts of classifiers.

We present classification task results for the six best and most popular classification algorithms. These algorithms/classifiers are support vector machine (SVM), k-nearest neighbors (KNN), decision tree (DT), ensemble learning, naive Bayes (NB), and discriminant analysis (DA). The following metrics were used to assess the performance of these classifiers. Some of these metrics are commonly used to evaluate the performance of an ML/DL model that has been trained. We utilized accuracy, sensitivity, specificity, precision, and F-score between these metrics. The most essential metric is accuracy, which indicates how many samples are properly categorized out of all the samples. It is commonly stated as the proportion of true positives (TPs) to true negatives (TNs) divided by the number of TPs, TNs, false positives (FPs), and false negatives (FNs) (FNs). A TP or TN is a data sample that is correctly classified as true or false by the algorithm. An FP or FN, on the other hand, is a data sample that the algorithm incorrectly classifies. This metric is represented by Equation (Equation 9).
(9)$$\begin{array}{c} \begin{array}{c} Accuracy\;=\;\frac{TP+TN}{TP+TN+FP+FN} \end{array} \end{array}$$

The recall metric is another name for the sensitivity metric. It is defined as the number of correctly categorized positive samples, implying how many samples of the positive classifications are correctly identified. It is given by Equation (Equation 10).
(10)$$\begin{array}{c} \begin{array}{c} Sensitivity/Recall\;=\;\frac{TP}{TP+FN} \end{array} \end{array}$$

The specificity of the given class pertains to the forecast of the possibility of a negative label turning true. It can be represented as in Equation (Equation 11).
(11)$$\begin{array}{c} \begin{array}{c} Specificity\;=\;\frac{TN}{TN+FP}\; \end{array} \end{array}$$

Equation (Equation 12) gives precision as an amount of TP divided by the number of TPs plus the number of FPs. This metric is all about regularity. In other words, it assesses the algorithm’s prediction accuracy. It determines how exact a model is based on what is predicted to be positive and how many of them are truly positive.
(12)$$\begin{array}{c} \begin{array}{c} Precision\;=\;\frac{TP}{TP+FP} \end{array} \end{array}$$

Finally, Equation (Equation 13), which is described as the relative average of precision and recall, provides the F-score. It is dependent on favorable class evaluations. This parameter’s high value indicates that the model works better in the positive class.
(13)$$\begin{array}{c} \begin{array}{c} F-score\;=\;2\times (\frac{Precision\times Recall}{Precision+Recall})\; \end{array} \end{array}$$

Table 1 shows the results obtained using the six classification methods stated above depending on the performance metrics of Robostar’s experimental setup. It delivers that the overall performance metrics for the Robostar fault detection show extremely promising results and the specified method works well for fault classification.

It is important to note that there are two classes in this situation: *normal* and *faulty*. The SVM classifier yielded the best accuracy. The NB had the lowest accuracy score. The SVM’s overall performance in distinguishing between positive and negative observations was likewise quite good. All of the findings given in the paper were achieved using 5-fold cross-validation to avoid overfitting in the classification model. The dataset is split into five distinct sets of disjoints, each comprising observations from each of the classes in 5-fold cross-validations. During the training process, one set is kept aside for testing while the others are utilized for training. This procedure is repeated for all folds, and the average accuracy is computed based on the testing results. To validate the results, we performed the training of the models for the repetition of 10 with a different range of random numbers. In the end, the average performance score was calculated by taking the average of the performance metrics for each classifier. The score shows that, regardless of the type of the classifier, on average, 96.26% accuracy, 96.17% sensitivity, 96.30% specificity, 96.19% precision, and 96.16% F-score can be achieved using the wavelet scattering network. Furthermore, the parallel coordinate plot of the 10 most prominent features are presented in Figure 20 and Figure 21, highlighting the two-class confusion matrix for the highest scoring classifier, i.e., SVM.

Figure 20 depicts the connection between the retrieved features and the predictors for class separation. The orange lines represent the *normal* class, whereas blue lines represent the *faulty* class. The wrongly categorized classes are shown by the light orange and blue lines. The standard deviation among the features is also presented along the y-axis of the plot to observe the difference in the data features. It can be seen that the classifier performs better when the standard deviation between the feature values of the classes is considerably different. Only 1 of the 800 samples/observations was incorrectly classified as *normal* when it was actually *faulty*. This is also supported by the model’s confusion matrix (shown in Figure 21).

Table 2 shows the results obtained using the six comparable classification methods based on the performance metrics for the Hyundai robot. In this case, three classes are used: *normal*, *faulty*, and *faulty aged*. The ensemble learning classification method achieved the highest accuracy, followed by the DA and SVM. With 88.1% accuracy, 88.006% sensitivity, 88.14% specificity, 88.029% precision, and 87.99% F-score, ensemble learning surpassed the DT and NB in the classification of faults associated with the rotate vector (RV) reducer for the Hyundai robot. The results demonstrate that selecting a suitable classification method based on criteria such as input data type, the number of classes, data quality, and data quantity is a critical step in any classification task. There has been no particular research on the selection of classification algorithms based on the aforementioned characteristics up to this point. As a result, to obtain greater classification accuracies, alternative sets of classifiers must be evaluated. It should also be noted that the NB and DT had the lowest accuracy in both the Robostar and Hyundai robot fault classification procedures. One of the reasons for this is because these two classification algorithms are built on a distinct set of ideas in contrast to the SVM, KNN, and DA. In terms of class clustering, SVM and KNN are virtually identical. Furthermore, as compared to the Robostar fault classification, the classification accuracy for the three-class classification task was relatively poor. This is because the inclusion of a third class in the classification problem caused substantial confusion among the *normal*, *faulty*, and *faulty aged* classes. This can be seen in Figure 22 and Figure 23, which exhibit a parallel coordinate plot of the ten most prominent features as well as a confusion matrix. The temporal and frequency domain differences in the Hyundai robot’s current signal for different classes were likewise not significant enough. A similar trend can be seen in the wavelet scattering network-extracted data features. There are several misclassified classes, particularly between *normal* and *faulty*, followed by the *normal* and *faulty aged* combination. In terms of computational cost, the results are still considerably more practical than other approaches that employ an extra 2D Scalogram image-based signal representation with a very deep layered network. This was verified by using a similar dataset for the Hyundai robot with a Scalogram image representation plus a CNN model. The highest accuracy achieved was no more than 80%. The most significant advantage of the wavelet scattering network is its computational simplicity, which allows for faster and shorter response times in real-time applications, particularly in applications such as this one. This approach is non-expansive if the wavelets and network architecture are chosen correctly. Energy dissipates as it passes down through the network. The energy of the *m*th-order scattering coefficients rapidly coheres to 0 as the rank of the network’s levels/layers rises [42]. Energy dissipation has a useful value. It reduces the number of wavelet filter banks in the network compared to a standard deep neural network while retaining as much signal energy as possible. The results demonstrate that by adding the deep scattering spectrum into a PHM application with real-time implementation, domain knowledge, and 1D signal-based data-majored environment, the computing complexity of the PHM systems can decrease significantly. Furthermore, in our future study, we plan to investigate the plausible solution and enhancement of the classification capacity for complex problems such as the Hyundai robot to develop a strategy that utilizes the wavelet scattering network’s fundamental advantages.

## 4. Conclusions

The use of a deep scattering spectrum (DSS) for 1D signal domains for the application of PHM systems, rather than traditional techniques such as scalogram conversion and image-based deep neural networks, has numerous advantages. Signals, being another type of information with its own set of properties, necessitate the use of techniques capable of extracting meaningful information from its non-stationary frequency-domain complicated space. Wavelets and other transformation techniques have been created and utilized for this purpose throughout the decades. The DSS blends the power of these signal-processing techniques with the deep learning paradigm. Surprisingly, the techniques involved are identical to those used in deep neural networks, but with the added benefit of more accurate information extraction. This research focuses on the implementation of the core concept of DSS, which is a wavelet scattering network for the fault classification of the mechanical components of various industrial robots. The utilization of an electric current signal for such fault classifications becomes a difficult task when an indirect electromechanical relationship of electric motors with their constituent mechanical components is exploited as the foundation for fault pattern recognition. Regardless of the intricacy, the results demonstrate that the DSS has an enormous potential in addressing fault classification issues under various circumstances. Furthermore, the future study would concentrate on the DSS’s enhancements, limits, and disadvantages for PHM applications, since we intend to integrate more faults linked to other mechanical components with a generalized feature space to construct a system-level real-time PHM system.

## Figures and Tables

**Figure 1 sensors-22-09064-f001:**
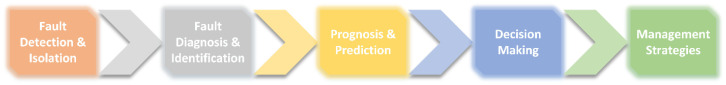
Basic tasks involved in a typical PHM system.

**Figure 2 sensors-22-09064-f002:**
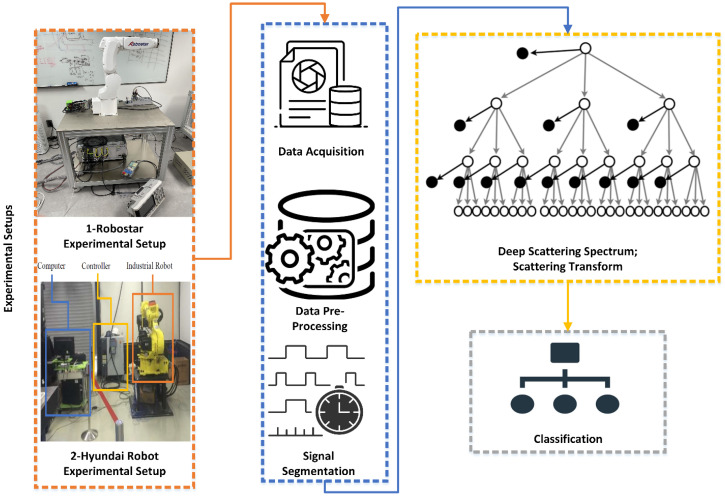
Basic overview of the workflow.

**Figure 3 sensors-22-09064-f003:**
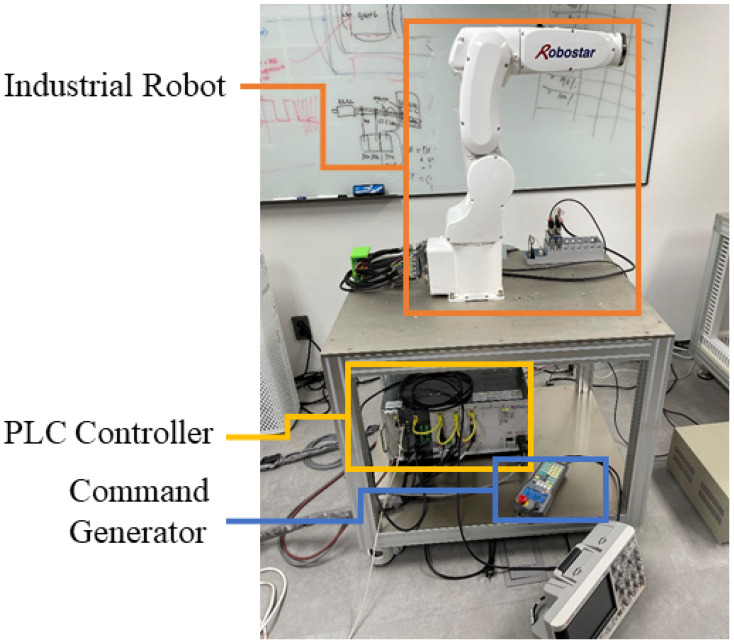
Experimental Setup for Robostar R004.

**Figure 4 sensors-22-09064-f004:**
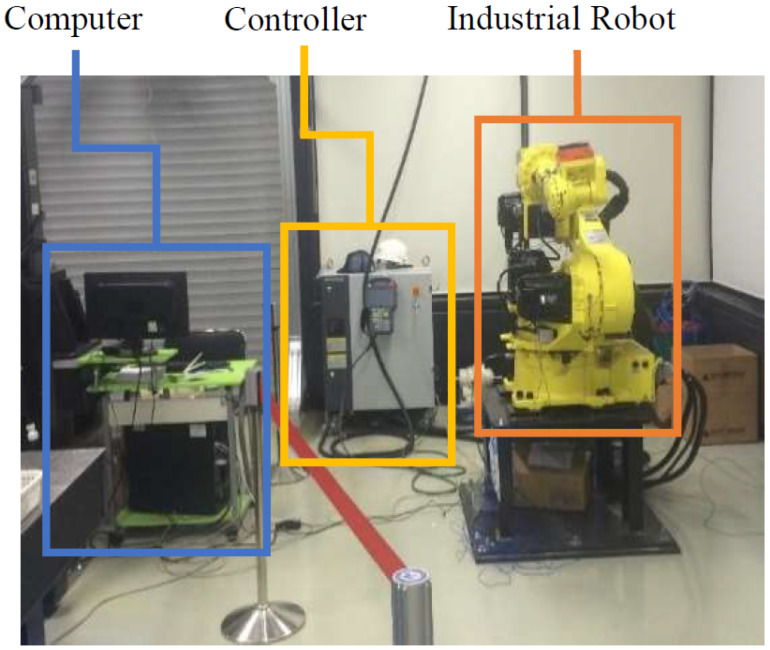
Experimental Setup for Hyundai Robot YS080.

**Figure 5 sensors-22-09064-f005:**
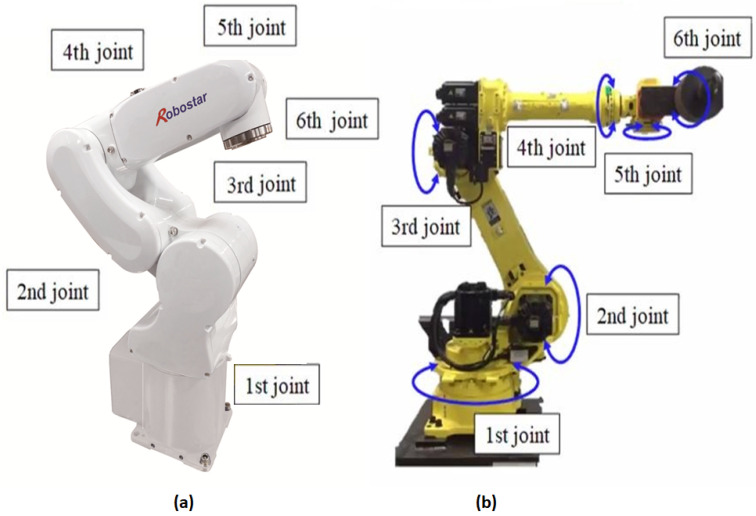
(**a**) Robostar R004 and (**b**) Hyundai Robot YS080.

**Figure 6 sensors-22-09064-f006:**
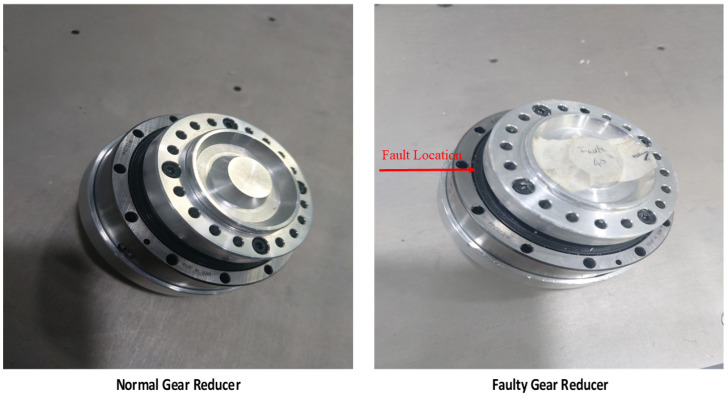
Strain wave gear reducer for Robostar Axis 3: normal and faulty.

**Figure 7 sensors-22-09064-f007:**
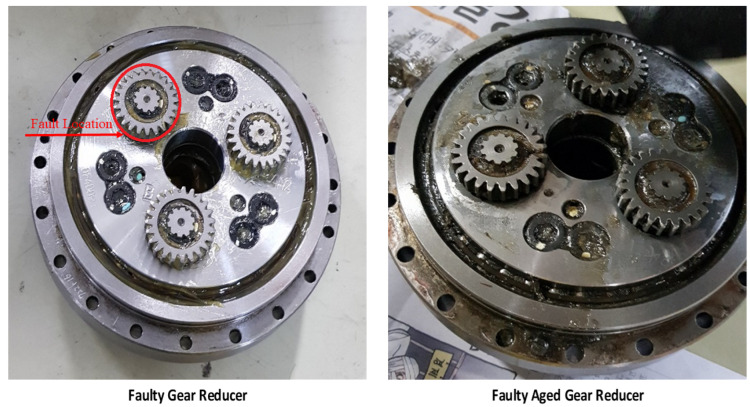
RV reducer gear for Hyundai Robot: faulty and faulty aged.

**Figure 8 sensors-22-09064-f008:**
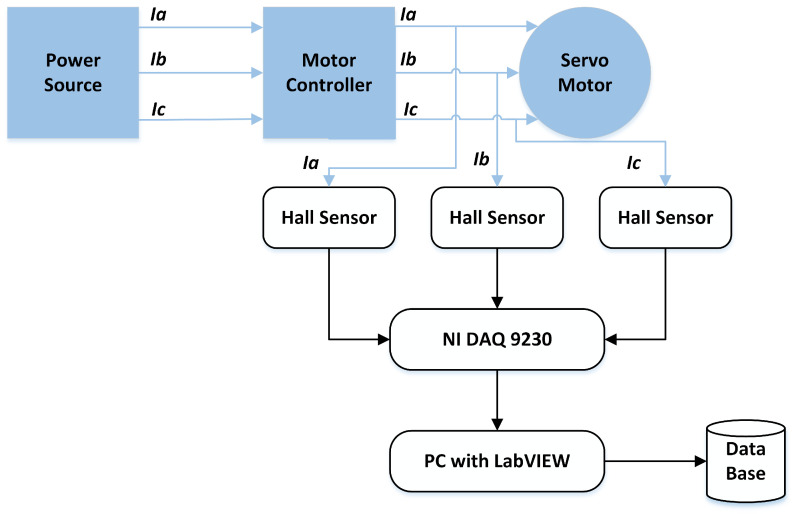
Basic overview of the data acquisition process.

**Figure 9 sensors-22-09064-f009:**
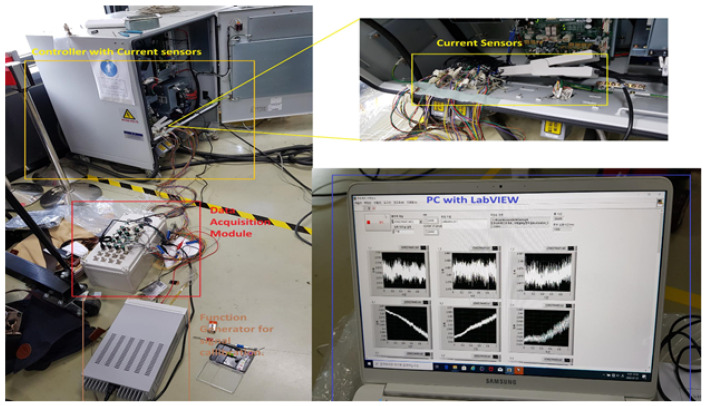
Data acquisition system for the Hyundai robot.

**Figure 10 sensors-22-09064-f010:**
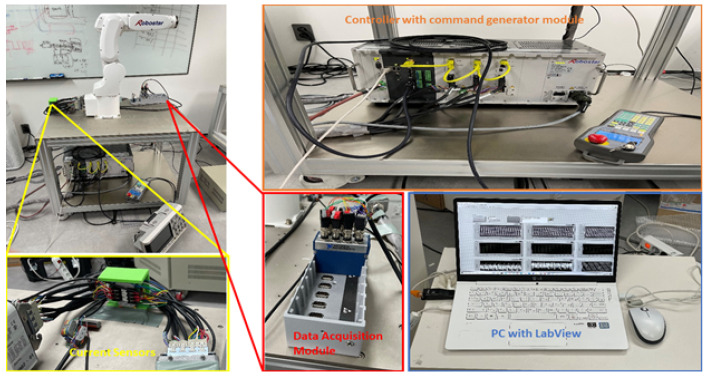
Data Acquisition system for Robostar.

**Figure 11 sensors-22-09064-f011:**
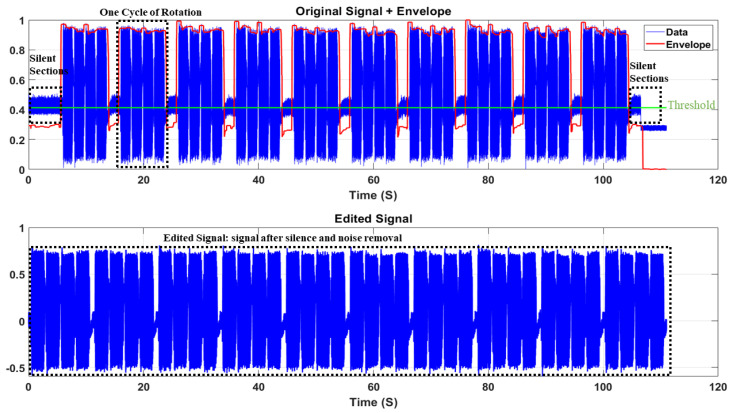
An example of the raw recorded current signal for Robostar at 10% the speed of rotation.

**Figure 12 sensors-22-09064-f012:**
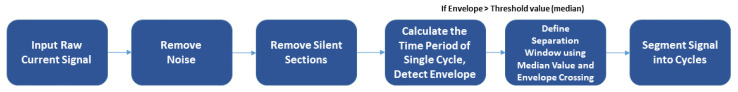
General block architecture of the segmentation method.

**Figure 13 sensors-22-09064-f013:**
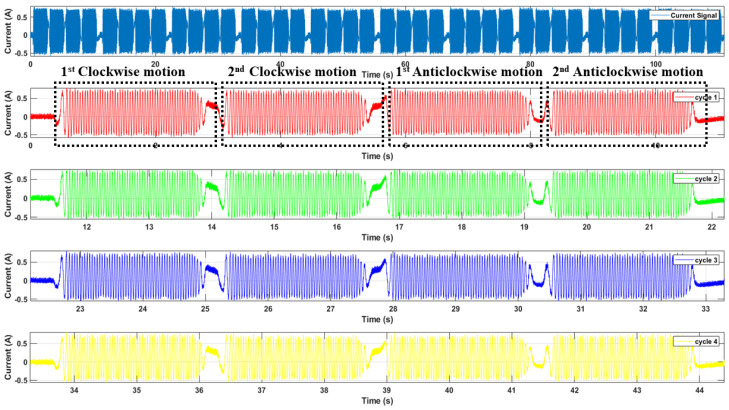
An example of segmented cycles for the signal shown in Figure 11.

**Figure 14 sensors-22-09064-f014:**
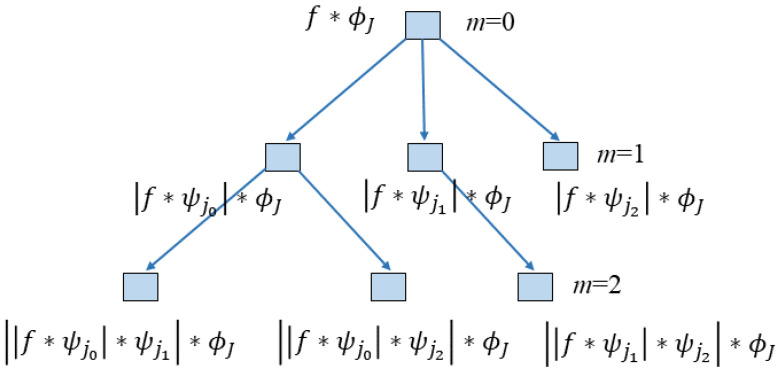
Wavelet scattering transform with two levels architecture for the 1D signal.

**Figure 15 sensors-22-09064-f015:**
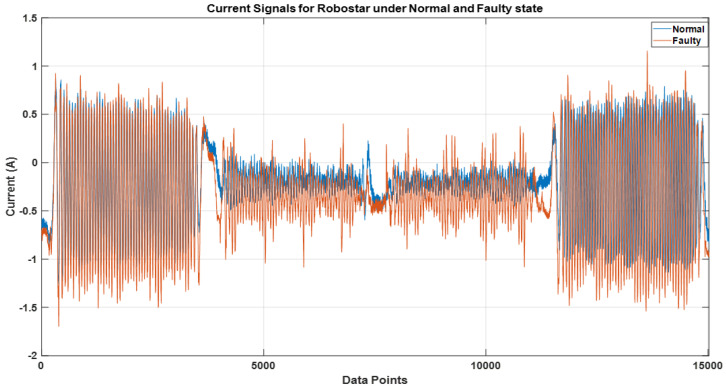
Electric current signal for one cycle of rotation for Robostar.

**Figure 16 sensors-22-09064-f016:**
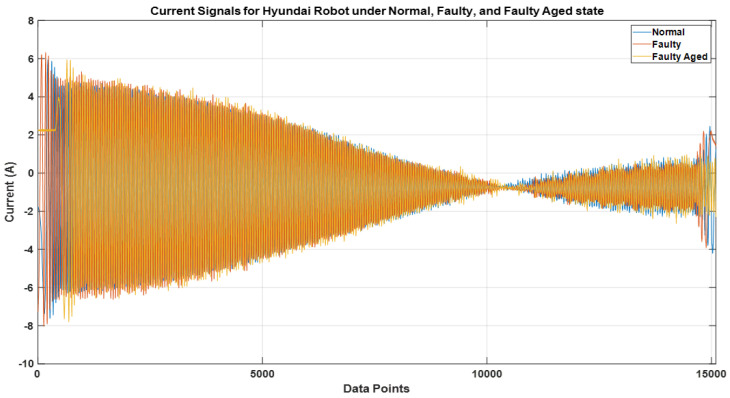
Electric current signal for one cycle of rotation for the Hyundai robot.

**Figure 17 sensors-22-09064-f017:**
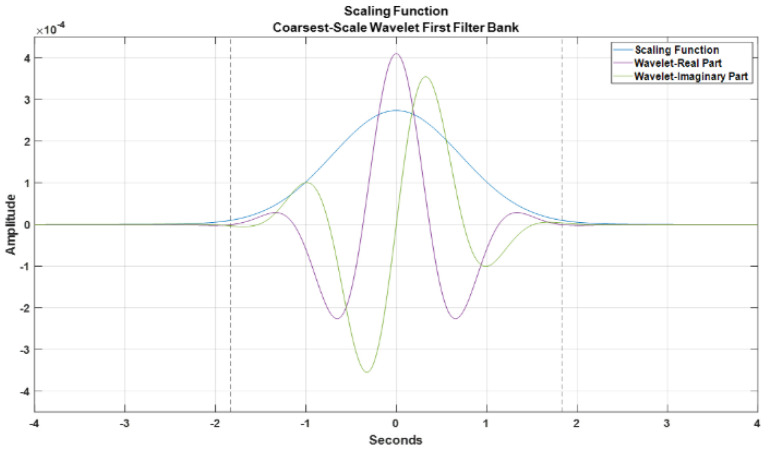
Scaling function and the coarsest-scale wavelet plot for the wavelet scattering network.

**Figure 18 sensors-22-09064-f018:**
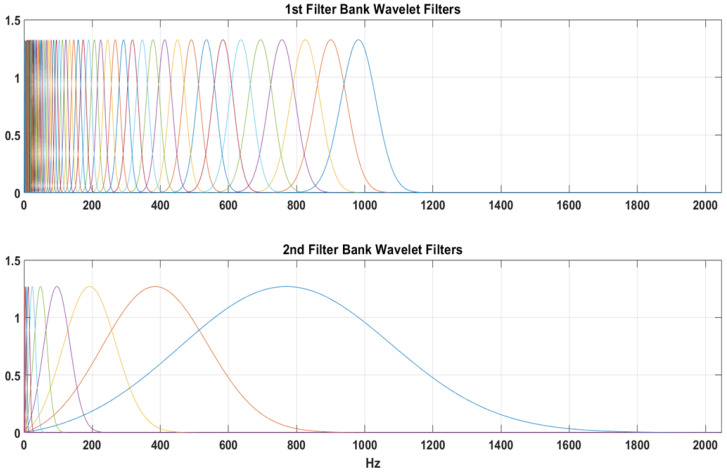
Wavelet scattering network’s filter banks at each layer.

**Figure 19 sensors-22-09064-f019:**
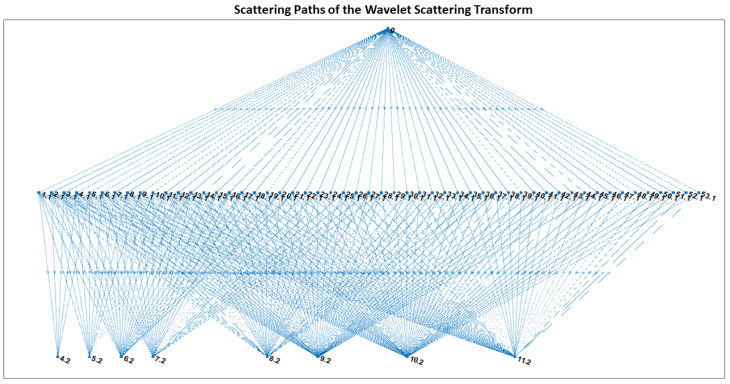
The wavelet scattering network’s paths based on the invariance scale, signal input length, and sampling frequency.

**Figure 20 sensors-22-09064-f020:**
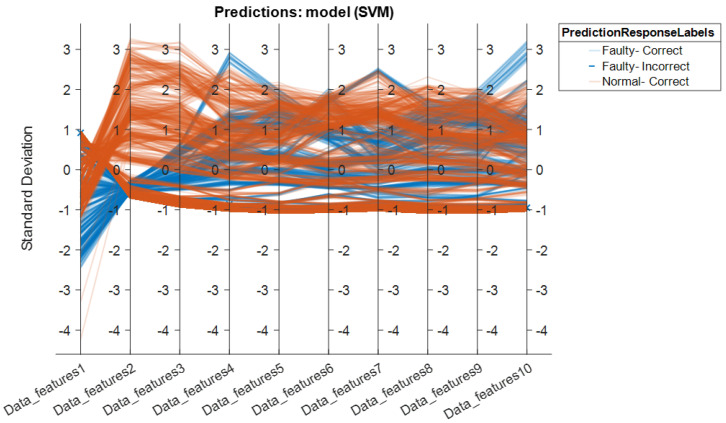
Parallel coordinate plot for the highest-scoring classifier (SVM) for Robostar fault classifications.

**Figure 21 sensors-22-09064-f021:**
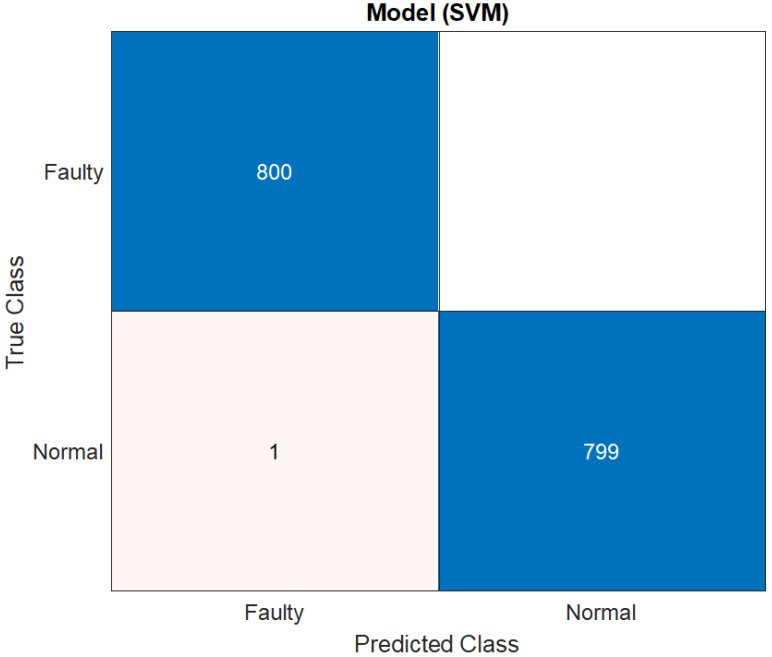
Confusion matrix for the highest-scoring classifier (SVM) for Robostar fault classification.

**Figure 22 sensors-22-09064-f022:**
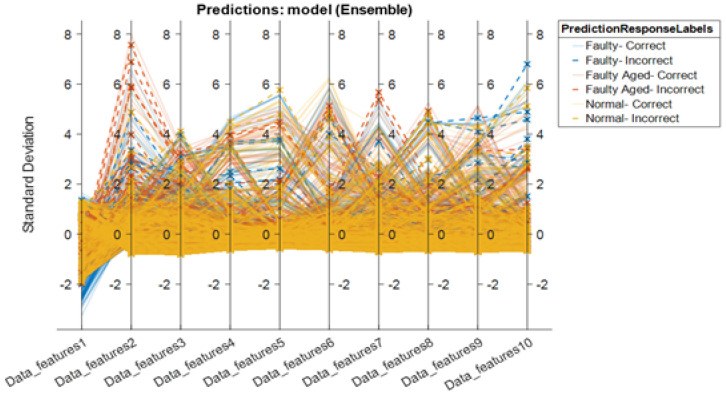
Parallel coordinate plot for the highest-scoring classifier (Ensemble learning) for Hyundai robot fault classification.

**Figure 23 sensors-22-09064-f023:**
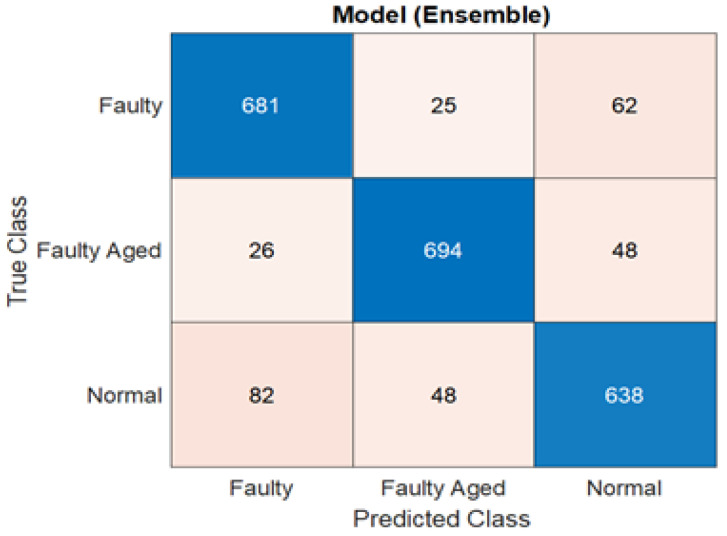
Confusion matrix for the highest-scoring classifier (Ensemble learning) for Hyundai robot fault classification.

**Table 1 sensors-22-09064-t001:** Classifiers performance score for Robostar fault detection.

Robostar Fault Detection		Metrics				
**Classifier**	**Number of Classes**	**Accuracy (%)**	**Sensitivity (%)**	**Specificity (%)**	**Precision (%)**	**F-Score (%)**
SVM		99.9	99.806	99.943	99.829	99.796
Decision Tree		99	98.906	99.043	98.929	98.896
Ensemble		99.6	99.506	99.643	99.529	99.496
KNN	2	99.4	99.306	99.443	99.329	99.296
Discriminant Analysis		97.7	97.606	97.743	97.629	97.596
Naïve Bayes		82	81.906	82.043	81.929	81.896
Average Performance Score		96.26	96.17	96.30	96.19	96.16

**Table 2 sensors-22-09064-t002:** Classifiers performance score for Hundai robot fault detection.

Hyundai Robot Fault Detection		Metrics				
**Classifier**	**Number of Classes**	**Accuracy (%)**	**Sensitivity (%)**	**Speciftcity (%)**	**Precision (%)**	**F-Score (%)**
Ensemble		88.1	88.006	88.143	88.029	87.996
Discriminant Analysis		85.6	85.506	85.643	85.529	85.496
SVM		83.2	83.106	83.243	83.129	83.096
KNN	3	80.3	80.206	80.343	80.229	80.196
Decision Tree		68.3	68.206	68.343	68.229	68.196
Naive Bayes		48.9	48.806	48.943	48.829	48.796
Average Performance Score		75.733	75.639	75.776	75.662	75.629

## Data Availability

The data are not publicly available due to privacy restrictions.

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
