# Peer review of "Deep Scattering Spectrum Germaneness for Fault Detection and Diagnosis for Component-Level Prognostics and Health Management (PHM)"

_sensors, 2022, doi:10.3390/s22239064_

Round 1

Reviewer 1 Report

I congratulate the authors for the originality of the research. No corrections or suggestions are necessary to enhance the comprehensiveness of the work.

Author Response

Thank you for considering the research study original and interesting. 

Reviewer 2 Report

1.      The writing quality of the abstract is not good, please rewrite, considering whether the logic is reasonable, for example, What is your topic/ problem? What's your solution, and How your solution works? How about the results?

2.      The title of your paper is hard to understand, please simplify and highlight the main points.

3.      Your main work on your manuscript is about deep scattering spectrum germaneness to fault detection and diagnosis, It's not the domain of this journal.

Author Response

Thank you for your comments and suggestions. Please find the attached document pertaining to the responses. 

Reviewer 3 Report

This manuscript proposed a deep scattering spectrum method to fault detection for robot PHM. the subject is hot in the mechanical fault diagnosis. The target is novel, however, there are some issues that need to be addressed before published:

(1) The abstract need to be improved to focus on the contribution of the paper.

(2) In the PHM area, there are many papers published, it is suggested to added the latest deep learning- based fault diagnosis articles in the Introduction, such as doi.org/10.1016/j.measurement.2021.110622; doi.org/10.1016/j.measurement.2022.111997.

(3) What is the advantage of the proposed DSS method to other current methods ? Please compare in experimental investigation.

(4) Which wavelet basis function is used in the paper ?

(5) It is suggested to mark the fault location with a circle in Figure 7.

(6) Is it more appropriate to change the paper title to "Deep Scattering Spectrum germaneness to Fault Detection and Diagnosis for Component-level Robot Prognostics and Health Management" ? 

Author Response

(The authors gave the same response as above.)

Round 2

Reviewer 2 Report

The authors have revised the  manuscript well and I recommend it for publication.

Reviewer 3 Report

Authors improved the manuscript according to the reviews' comments, I think it can be published in Sensors.